# Soft Prompt-tuning for Short Text Classification via Internal Knowledge Expansion

## Abstract

The last decades have witnessed a vast amount of interest and research on short texts. The limited contextual information, feature sparsity, and semantic ambiguity accentuate the main challenges of short text classification. Recently, pre-trained language models (PLMs) have achieved tremendous success in various downstream Natural Language Processing (NLP) tasks including short text classification. However, most of the existing methods rely on the external expansion from the open knowledge base to address the inherent limitations of short texts, which not only inevitably incur a time-consuming query process and the omissions and biases in noise, but also rely on the high-quality open knowledge base and cannot be applied in some real-world off-line scenarios. In this paper, we propose a novel Soft Prompt-tuning method for short text classification via Internal Knowledge Expansion (SPIE). Our method stems from the recent success of prompt-tuning and extracts knowledge from the training dataset itself. We conduct hierarchically cluster and optimization strategies to fine-tune the obtained expansion words for the verbalizer in prompt-tuning. Furthermore, we employ soft prompt-tuning to avoid bias introduced by hand-crafted templates and improve the overall performance of the model. Despite internal expanding knowledge, experimental results demonstrate that our method even outperforms the methods that introduced external knowledge with much less computational time on four well-known benchmarks.

## 1 Introduction

With the explosive growth of online platforms and internet content in recent decades, short text has significantly increased at unprecedented rates. This surge encompasses diverse forms of information, including news headlines, search snippets, social media posts, and so on. The rapid development of short texts has also highlighted the challenges in extracting meaningful information from these texts, given their inherent limitations in terms of limited contextual information, sparse features, and semantic ambiguity (Zhou et al., 2022a). Recently, short text classification has garnered substantial attention across multiple disciplines, with advancements in this field yielding profound impacts on various practical applications (Qiang et al., 2020b).

Recent breakthroughs in Pre-trained Language Models (PLMs) (Han et al., 2021a) have cemented their position as indispensable tools for a wide array of downstream Natural Language Processing (NLP) tasks, including short text classification (Kowsari et al., 2019). These models are pre-trained on extensive, unlabeled datasets using unsupervised learning techniques, enabling them to acquire rich contextual representations while significantly reducing the dependency on large-scale annotated data (Ye et al., 2020). By fine-tuning PLMs on a small amount of task-specific data, researchers can effectively leverage the extensive linguistic and semantic knowledge encoded within these models, facilitating their application to diverse NLP tasks, such as lexical simplification (Qiang et al., 2020a), machine translation (Conneau & Lample, 2019), and question answering (Tu et al., 2020). The exceptional performance achieved through fine-tuned PLMs has rendered the process of training models from scratch largely redundant (Han et al., 2021b).

Although fine-tuned PLMs have achieved remarkable success across various tasks, the substantial gap between the objective function during pre-training and the fine-tuning process limits their ability to fully exploit the rich knowledge embedded within these models. Inspired by GPT-3 (Brown et al., 2020), prompt-

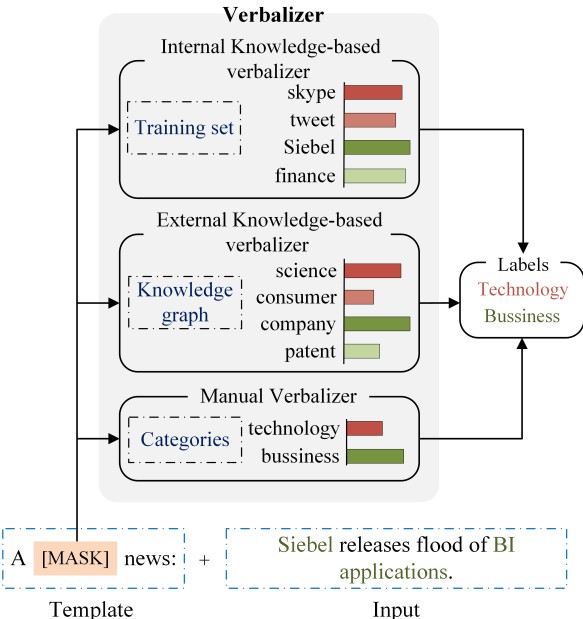

Figure 1: Illustration of three verbalizer construction methods.

tuning addresses this limitation by reformulating traditional NLP tasks into cloze-style tasks. This approach embeds the input into a natural language template and leverages a masked language model for prediction, demonstrating exceptional performance in few-shot and zero-shot learning scenarios. For instance, when classifying the sentence $x = "Siebel\ releases\ flood\ of\ BI\ applications."$ into the category "$Business$", a template can be constructed as "$x,\ the\ topic\ is\ about\ [MASK]$." The prompt-tuning model then predicts the probability of the word "$Business$" filling the $[MASK]$. Furthermore, a mapping mechanism, referred to as the verbalizer, aligns label words (e.g., "business", "market", "economics") with their corresponding categories (e.g., "Business"). This mapping effectively bridges the semantic gap between the textual representation space and the label space. Empirical studies have demonstrated that this strategy significantly enhances performance in short text classification tasks (Schick et al., 2020; Ma et al., 2024).

Existing approaches to short text classification can generally be categorized into two primary types: sole-source-based methods and external-resource-based methods. Sole-source-based methods aim to expand the feature space by relying exclusively on rules or statistical information derived from the short text itself (Cui et al., 2022; Ai et al., 2024). However, these approaches often face challenges related to feature sparsity, which can significantly impair classification performance. In contrast, external-resource-based methods incorporate additional knowledge to enhance classification accuracy (Chen et al., 2019; Gao et al., 2023). Among these, prompt-tuning techniques have emerged as a prominent focus for short text classification. These methods can be further subdivided into three categories: manual verbalizers, external-knowledge-based verbalizers, and internal-knowledge-based verbalizers. A visual representation of these categories is provided in Figure 1.

Although external-resource-based methods have demonstrated commendable performance, two significant challenges persist, hindering the advancement of short text classification. The first challenge lies in their heavy dependence on the quality of open knowledge graphs, which limits their applicability in certain real-world offline scenarios. In domains with incomplete or specialized knowledge bases, this reliance can result in performance degradation. The second challenge concerns the inherent inefficiency of external knowledge expansion. This process often involves time-intensive queries and introduces the risk of omissions and biases due to noise. Compared to internal-knowledge-based methods, external-resource-based approaches not only introduce noise alongside knowledge but also incur substantial query times, which are often constrained by network availability and efficiency.

To address these challenges, we propose an innovative approach: Soft Prompt-tuning for short text classification via Internal Knowledge Expansion (SPIE), leveraging the strengths of knowledgeable prompt-tuning models. This method tackles the issue of feature sparsity in short texts while avoiding the noise and high computational complexity associated with external knowledge integration. Our approach constructs a verbalizer by extracting nouns and adjectives directly from the original training dataset, thereby expanding conceptual information from the short text itself rather than relying on external knowledge bases. Additionally, soft prompts are employed to enrich contextual information, enabling dynamic adjustments during training to optimize prompts for specific text types. This strategy not only mitigates biases introduced by manual templates but also enhances the model's generalization ability. Extensive experimental results demonstrate that SPIE not only achieves superior performance compared to methods incorporating external knowledge but also significantly reduces computational overhead, making it an efficient and effective solution for short text classification.

Our approach enhances the features of short texts without introducing external noise, simplifies prompt construction, and facilitates obtaining the optimal prompt. The main contributions of this paper are summarized as follows::

- Internal Knowledge Expansion: We extract internal knowledge, such as nouns or adjectives, from the training dataset and expand label vocabulary through clustering methods. Our method avoids introducing noise from external knowledge and enhances model performance.

- Soft Template Construction: We employ soft templates to construct inputs for the prompt-tuning model, reducing biases introduced by manual template construction and enhancing the model's generalization capability.

- Extensive experimental validation: Extensive experiments demonstrate that our method surpasses the performance of SOTA methods. Despite internal expanding knowledge, experimental results demonstrate that our method even outperforms the methods that introduced external knowledge with much less computational time on four well-known benchmarks.

## 2 Related Work

Over the last decades, short texts have garnered significant attention due to their critical role in numerous real-world applications. In recent years, PLMs and prompt-tuning methods have shown remarkable performance in various downstream NLP tasks. In this section, we review the literature on short text classification, prompt-tuning, and the development of verbalizers.

### 2.1 Short Text Classification

The short text has garnered significant attention and research in recent decades, playing a vital role in various practical applications, including sentiment analysis (Song et al., 2020), dialogue systems (Forgues et al., 2014), and user intent understanding (Blackman Sphaier & Paes, 2022). Short text classification aims to handle very short texts, typically not exceeding 100 characters, such as blog content, online comments, news headlines, and more.

Existing methods for short text classification can be broadly categorized into two main classes: sole-source-based and external-resource-based methods. The sole-source-based methods mainly depend on the text itself to identify the underlying factors that explain variations in the short text. In contrast, external-resource-based methods leverage external sources, such as open Knowledge Graphs, to augment knowledge for improved representation learning on short texts.

The sole-source-based methods focus on utilizing large-scale data themselves, training models through machine learning and deep learning techniques, without relying on external knowledge bases. For example, Kim et al. proposed a CNN-based method for short text classification, which proves especially suitable for tasks involving sentence-level classification (Kim, 2014). Hu et al. proposed a flexible framework, the Heterogeneous Information Network (HIN), specifically designed to represent short texts. This framework

integrates various types of supplementary information and captures their interrelationships, addressing the challenge of semantic sparsity (Linmei et al., 2019). Hao et al. designed a CNN model enhanced by a mutual attention mechanism, which leverages both character-level and word-level features to improve the model's ability to capture fine-grained information (Hao et al., 2020). Zhou et al. put forward a multi-channel convolutional framework that simultaneously captures local and global semantic patterns. A semantic expansion mechanism, based on a rapid clustering algorithm, is employed to enrich the feature set of short texts (Zhou et al., 2022b). Tian et al. proposed a multi-label classification method for social short texts, combining contrastive learning with an enhanced KNN approach. This method improves feature representation by leveraging knowledge from existing samples, thereby alleviating issues related to semantic sparsity (Tian et al., 2024).

Although sole-source-based methods have been successful, some studies suggest that these approaches may encounter issues related to feature sparsity. In contrast, external-resource-based methods incorporate additional auxiliary knowledge or domain-specific expertise to enrich the feature set of short texts. Conventional external-resource-based methods include expanding vocabularies or disambiguating word meanings using external knowledge bases, as well as embedding information from external domain knowledge bases into feature representations. For example, Liu et al. proposed a hierarchical attention model that combines CNN and Temporal Convolutional Networks, and enhances semantic representation by incorporating the external knowledge base Probase (Liu et al., 2022b). Xu et al. introduced a hybrid method that integrates context-specific knowledge with a CNN to improve short text classification (Xu & Cai, 2019). Yang et al. introduced a method that utilizes a heterogeneous graph convolutional network to model the relationships between words and their associated concepts. This method utilizes graph convolutional networks to learn representations by capturing the interactions between these elements (Yang et al., 2023). Chen et al. proposed an improved prompt tuning method that enhances the semantic depth of short texts by incorporating external knowledge into the soft prompt template and leveraging a separating soft verbalizer (Chen et al., 2024). Despite these methods can deliver impressive performance, they depend heavily on large-scale open knowledge bases. This reliance introduces several challenges, including potential omissions, time-consuming query processes and biases due to noise, and dependence on high-quality knowledge sources. As a result, these methods may not be applicable in certain real-world offline scenarios.

## 2.2 Prompt-tuning

While fine-tuning methods for PLMs have yielded promising results, recent studies highlight a critical challenge: the substantial gap between the objective functions used in pre-training and those in fine-tuning. This gap limits the full exploitation of the knowledge embedded in PLMs. Inspired by GPT-3 (Brown et al., 2020), prompt-tuning transforms the original natural language processing task into a cloze-style task by embedding input information into a natural language template and adapting a masked language model. Prompt-tuning has demonstrated exceptional performance across various downstream NLP tasks, including data augmentation (Wang et al., 2022), relation extraction (Chen et al., 2022), text generation (Zhang & Song, 2022), and text classification (Zhu et al., 2022), especially in scenarios with limited data samples.

The primary components of prompt-tuning consist of a template and a set of label words. The template serves to describe the background information of the task, while the label words denote the high-probability vocabulary predicted by PLMs within a specific context. In the design of manual templates, the discrete textual prompts are hand-crafted by human experts and incorporate prior knowledge while remaining unchanged throughout the training process. For example, Han et al. proposed a relation classification method that generates multiple sub-prompts based on logic rules (Han et al., 2021b). Furthermore, Ding et al. proposed to address the challenge of fine-grained entity typing by incorporating manually designed templates within the framework of masked language modeling (Ding et al., 2021). Gu et al. introduced an innovative prompt pre-training strategy tailored for few-shot learning, which optimizes the pre-training task to achieve notable improvements while significantly reducing the reliance on large-scale datasets (Gu et al., 2021).

While manual templates have demonstrated effectiveness across numerous NLP tasks, their construction is often labor-intensive and time-consuming. Moreover, poorly designed templates can result in a degradation of model performance. To this end, recent research has focused on the developments of soft template generation in prompt-tuning. Soft templates are continuous prompts, commonly represented as vectors, that

are iteratively optimized during the training process to improve model performance. For instance, Lester et al. introduced Prompt Tuning, a method to adjust some parts of pre-trained NLP models. Instead of changing the whole model for different tasks, they only modify a specific subset of the prompt (Lester et al., 2021). Liu et al. developed an automated approach for generating prompts that customized templates for specific downstream tasks, seamlessly integrating flexible vector representations within the template structure (Liu et al., 2022a). This integration occurs through ongoing fine-tuning during the training process. Du et al. proposed a task-specific prompt template generation method that automatically produces optimal templates while integrating external knowledge to effectively enhance task performance (Du et al., 2024).

## 2.3 Verbalizer Construction

In the prompt-tuning, the verbalizer refers to the mapping of label words (e.g., "business", "market", "economics") to their corresponding categories (e.g., "BUSINESS"). This method has been demonstrated to effectively address the mismatch between the text and label spaces (Schick et al., 2020). The hand-crafted verbalizers have demonstrated strong performance in various downstream NLP tasks including text classification. For example, Schick et al. proposed a method that leveraged pre-defined pairs of cloze question patterns and manually engineered verbalizer to harness the inherent knowledge of PLMs for downstream tasks (Schick & Schütze, 2020). However, the construction of a manual verbalizer is notably susceptible to the influence of prior knowledge, which can introduce both bias and omissions during knowledge expansion.

As the development of a hand-crafted verbalizer requires prior knowledge and the performance is unstable, several methods for the automatic construction of verbalizers in prompt-tuning have been introduced. For instance, Wei et al. introduced a prototypical network that generated label-specific embeddings by encapsulating the semantic attributes of labels, effectively operating within the feature space to facilitate the construction of a prototypical prompt verbalizer (Wei et al., 2022). However, the method tends to expand synonyms of category names rather than a diverse and comprehensive set of label words. To reduce noise in the expanded label words of automatic verbalizers, several studies have explored the selection of relevant terms from external knowledge bases, as demonstrated by the Knowledgeable Prompt Tuning (KPT) method (Hu et al., 2021). While the method significantly enhances the semantic representation of labels, the extraction of numerous irrelevant words during the verbalizer construction phase makes it difficult to use directly, often resulting in suboptimal performance in downstream tasks. Building on the KPT approach, Ni et al. introduced the KPT++ method, which integrates two innovative mechanisms: Probability Distribution Refinement (PDR) and Prompt Grammar Refinement (PGR). These mechanisms are designed to further optimize the knowledge-enriched language model (Ni & Kao, 2023).

Unlike existing methods, our approach exclusively leverages internal knowledge derived from the training dataset. By utilizing soft prompt-tuning, we mitigate the potential biases introduced by manually crafted templates, thereby improving the overall performance of the model.

# 3 Methodology

## 3.1 Motivation

Recently, short texts have been posted at an unprecedented rate due to the rapid growth of the Internet. For example, search snippets often contain fewer than 20 words, while the content of tweets is limited to a maximum of 140 characters (Kenter & De Rijke, 2015). Due to the inherent limitations of short texts in the limited contextual information, feature sparsity, and semantic ambiguity, most existing short text classification methods introduce an external knowledge base to address these problems for achieving better performance. These methods seek to incorporate additional knowledge and have demonstrated notable success in short text classification. However, two significant challenges persist, hindering the further advancement of these methods.

The first problem is the high dependence on the quality of external knowledge bases, which not only makes these methods unsuitable for offline scenarios, but also leads to performance degradation in vertical domains without high-quality open knowledge. The second problem is noise and high time complexity during the

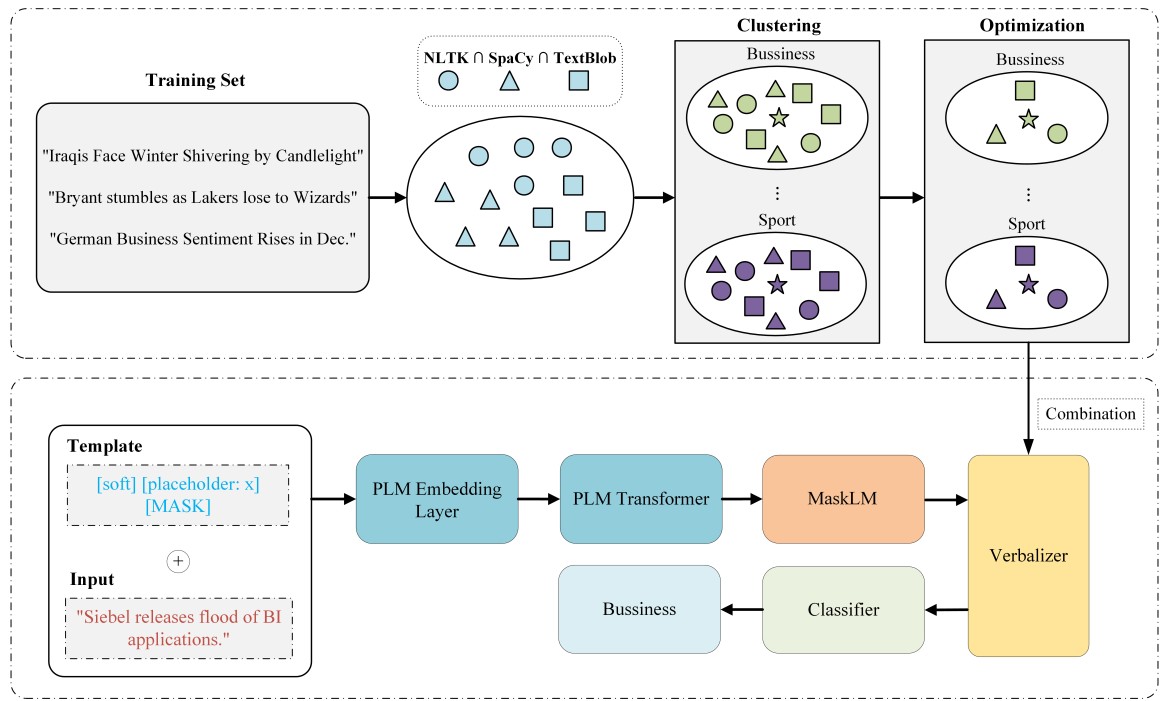

Figure 2: The overall framework of our SPIE. "[soft] [placeholder: x] [MASK]" is a soft template. The upper part describes the process of extracting extension words from the training dataset using three different tools. The bottom part pertains to the prompt-tuning model, in which the acquired extension words are ultimately utilized in the verbalizer for prompt-tuning.

process of external knowledge expansion. In contrast to the internal-knowledge-based methods, the external-knowledge-based methods introduce noise, such as biases and omissions, in the process of introducing knowledge, and conduct a time-consuming query depending on the network. For example, the Knowledgeable Prompt Tuning (KPT) method expands concepts in short texts by incorporating relevant vocabulary from external knowledge graphs (Hu et al., 2021). While the method can introduce a substantial amount of external knowledge, it also brings in a significant amount of noise that even cannot be addressed by the noise reduction models.

In this paper, our method expands knowledge from the short text itself by extracting adjectives or nouns from the internal training dataset, effectively avoiding the influence of external noise. Moreover, we optimize the extraction of internal vocabulary through specific strategies, building upon the recent success of prompt-tuning in our work. To avoid where the crafted template methods are time-consuming and labor-intensive, while automatic prompt generation methods cannot achieve satisfactory performance, we propose a soft template construction method, allowing for flexible prompt adjustments during training to obtain better prompts tailored to specific text types. This method not only reduces bias introduced by manual templates but also enhances the model's generalization capability.

## 3.2 Overall Architecture

Given a short text $x$ to be classified with the corresponding label $y$, the text classification task is formalized as the prediction $p(y|x)$. Our proposed method integrates recent advancements in prompt-tuning and introduces a novel approach to verbalizer construction, as illustrated in Figure2. Initially, nouns are extracted from the training dataset using three distinct tools, while adjectives are additionally extracted for sentiment-related datasets. Subsequently, the intersection of the vocabulary extensions generated by these tools is computed and clustered, with category-label words serving as cluster centers, thereby yielding extended words for

each category. Finally, these extended words are optimized using BERT, resulting in the final verbalizer construction.

### 3.3 Soft Prompt Construction

In the prompt-tuning, the input sentence is reformulated into a natural language template, transforming the short text classification task into a masked prediction task. For instance, in a classification task, where we need to determine if a short text $x =$ "*Siebel releases flood of BI applications.*" belongs to bussiness or technology, we first need to construct a manual template:

$$T = \{x, \text{ the topic is about } [MASK]. \} \tag{1}$$

Given an input $x = \{w_0, \ldots, w_n\}$ to be classified into a label $y \in Y$, the collection of label words will be denoted as $V_y = \{v_0, \ldots, v_n\}$, and $V_y$ will be mapped into a category with label $y$. In the pre-trained language model $M$, the probability of each word $v$ being filled in $[MASK]$ can be denoted as $p([MASK] = v \in V_y|x_p)$. Therefore, the classification problem of short text can be transformed into the problem of calculating the probability of label words, which is calculated as:

$$p(y \in Y|x) = p([MASK] = v \in V_y|x_p) \tag{2}$$

In the proposed SPIE, a soft template is utilized, in contrast to manual templates, to enable training within a continuous space of optimal prompts, which can be noted as:

$$T = \{[u_i], \ldots, x, \ldots, [u_n], [MASK]\} \tag{3}$$

where $u_i$ represents the $i^{th}$ learnable token, $x$ is a input text. The constructed prompt $T$ is then passed to a PLMs encoder, denoted as *Encoder*, to obtain the corresponding latent vector $h$:

$$\{h_i, \ldots, h_x, \ldots, h_n, h_{mask}\} = Encoder(T) \tag{4}$$

After extracting the hidden vectors of $T$, we employ a BiLSTM to capture the text's context and update the parameters related to learning the token $u_i$, as follows:

$$h_i = BiLSTM(h_i, h_x, h_n) \tag{5}$$

Finally, the optimized loss function is as followsequation 6:

$$h = \arg\min_{h_i} L(M(x, mask)) \tag{6}$$

where $M(x, mask)$ represents the output of PLMs $M$ on input data $x$ and mask, and $\arg\min_{h_i}$ denotes the value of variable $h_i$ that minimizes the loss function.

### 3.4 Verbalizer Construction

In the prompt-tuning, mapping label words to their corresponding categories through a verbalizer can effectively enhance the performance of downstream tasks. Unlike the external-resource-based methods, we select to extract specific nouns or adjectives from the training set of each dataset to expand label words. While part-of-speech tagging techniques are relatively mature, they may still have some errors. Therefore, we opt for three different part-of-speech tagging tools (NLTK[1], SpaCy[2], and TextBlob[3]) to assist us in extracting nouns or adjectives from the dataset in the experiments, exploring the effectiveness of different methods in vocabulary extraction.

---

[1] https://www.nltk.org/
[2] https://spacy.io/
[3] https://textblob.readthedocs.io/en/dev/

Firstly, for the training set $T = \{s_1, \ldots, s_i, \ldots, s_n\}$, where $s_i$ represents the $i^{th}$ sentence in the training set, we perform part-of-speech tagging using three different tools on all the words in the dataset. This is represented as:

$$N_{pos}(s_i) = \{(w_1, p_1), \ldots, (w_i, p_i), \ldots, (w_n, p_n)\} \tag{7}$$

$$S_{pos}(s_i) = \{(w_1, p_1), \ldots, (w_i, p_i), \ldots, (w_n, p_n)\} \tag{8}$$

$$T_{pos}(s_i) = \{(w_1, p_1), \ldots, (w_i, p_i), \ldots, (w_n, p_n)\} \tag{9}$$

where $N_{pos}(s_i)$, $S_{pos}(s_i)$ and $T_{pos}(s_i)$ represent part-of-speech tagging results obtained by three tools NLTK, SpaCy and TextBlob respectively, $w_i$ is the $i^{th}$ word in sentence $s_i$, and $p_i$ is the corresponding part-of-speech of $w_i$.

Then, we extract all the nouns from these tagged words. It is important to note that in sentiment datasets, such as Twitter, we extract adjectives. This process can be represented as:

$$\{n_1, \ldots, n_i, \ldots, n_n\} = Extract_N(U) \tag{10}$$

where $U = \{N_{pos}(s_i), S_{pos}(s_i), T_{pos}(s_i)\}$. Finally, we obtain three sets containing all the nouns or adjectives: $N_N$, $N_S$, and $N_T$.

Furthermore, as the part-of-speech tagging results from each tool may have some discrepancies, we select to take the intersection of these three vocabulary sets to determine the part-of-speech as accurately as possible. This process can be represented as:

$$N_{final} = N_N \cap N_S \cap N_T, \tag{11}$$

where $\cap$ represents the operation of taking the intersection.

Once we have obtained all the nouns from the training set, we use Hierarchical Clustering (HC) (Müllner, 2011) to consider the label words in the dataset as cluster centers and assign these vocabulary words to specific categories:

$$\{c_1, \ldots, c_i, \ldots, c_m\} = HC(N_{final}), \tag{12}$$

where $c_i$ is the final set of expanded words for the $i^{th}$ category.

However, this process may introduce some noisy words, potentially affecting the model's performance. Therefore, our method further adopts a BERT Prediction strategy to optimize the label words. To leverage the rich knowledge of pre-trained models, it is optimized based on the probability distribution of $[MASK]$ in the template. For example, in a template $T = $"$[placeholder : c]$, $which\ is\ related\ to\ [MASK]$", where $[placeholder : c]$ represents a specific category, the probability indicates the similarity with that category. We sort all the words based on these probabilities and select the top $N$ words as the final verbalizer expansion words.

### 3.5 Short Text Classification

The output of the PLM-Encoder is fed into the Transformer layer in the PLM:

$$h_{tm}, h_x, h_{mask} = M - Transformer(h) \tag{13}$$

Subsequently, as shown in equation 2, the context-aware vector $h_{mask}$ is processed through a multi-layer perceptron (MLP) to compute the probability distribution for each label word in the vocabulary. During training, the cross-entropy loss function is employed to update the model parameters $\theta$:

$$L = -\frac{1}{N_a} \sum_{i=1}^{N_a} w_i \log p\left(y_i^* \mid x_i\right) + \alpha \left\|\theta\right\|_1 + \beta L_{aux} \tag{14}$$

where $N_a$ denotes the number of samples, serving as a normalization factor to average the total loss. The summation $\sum_{i=1}^{N_a}$ aggregates the loss over all samples, with $w_i$ representing the weight assigned to the $i^{th}$

Table 1: Statistics of our dataset. "Avg.Len" denotes "the average number of words per text".

| Dataset | Class | Data Size | Avg.Len |
|---------|-------|-----------|---------|
| AG's News | 4 | 7600 | 7 |
| Snippets | 8 | 2280 | 18 |
| TMN News | 7 | 6521 | 8 |
| Twitter | 2 | 10000 | 11 |

sample. $y_i^*$ refers to the true label of the $i^{th}$ sample, and $p\left(y_i^* \mid x_i\right)$ is the predicted probability of the true label given the input $x_i$. The regularization term $\|\theta\|_1$, scaled by the coefficient $\alpha$, helps prevent overfitting by encouraging sparsity in the model parameters. Finally, $L_{aux}$, controlled by the weight $\beta$, represents the auxiliary task loss, which contributes to the overall optimization objective.

## 4 Experiments

In this section, we first introduce the details of the dataset and experimental setup. Subsequently, we provide the baselines employed for comparison. Then, we analyze our experimental results and conduct an ablation study. Finally, the influence of the time complexity, template, and hyper-parameters are explored, and then we present the visualization of expanded verbalizer weights.

### 4.1 Data Setting

The experiments are conducted on four well-known short text benchmark datasets, and Table 1 provides detailed statistics for each dataset. A specific description of each dataset is presented as follows:

**AG's News**[4]**:** The AG's News dataset is a well-established benchmark commonly used for categorizing news topics. It comprises 120,000 English articles sourced from more than 2,000 outlets and is categorized into four distinct topics: World, Sports, Business, and Science/Technology. The dataset is curated by selecting the largest four categories from the original AG's News collection. For experimental purposes, only the headlines of the articles are used as input, serving as brief textual representations.

**Snippets**[5]**:** The Snippets dataset includes three key elements: URL, title, and text description. It contains web search results generated by Google in response to user queries submitted to its search engine. Each search result is labeled with the same class as the query phrase that generated it, providing a direct link between the query and the associated snippet.

**TMN News**[6]**:** The TMN News dataset, similar to AG's News, consists of 32,000 English articles sourced from well-known newspaper websites. For consistency with AG's News experiments, only the article headlines are used as input texts.

**Twitter**[7]**:** The binary sentiment analysis dataset from NLTK consists of 10,000 tweets, evenly split between positive and negative sentiments, with 5,000 tweets in each category. Each record comprises a unique TweetID, the tweet's text (Text), and a sentiment label (Sentiment), where 0 represents negative and 1 indicates positive sentiment.

### 4.2 Baseline methods

To assess the performance of our proposed method, a comparative analysis is carried out against SOTA approaches, with the details outlined as follows.

---

[4]http://www.di.unipi.it/~gulli/AG_corpus_of_news_articles.html
[5]http://jwebpro.sourceforge.net/data-web-snippets.tar.gz
[6]http://acube.di.unipi.it/tmn-dataset/
[7]https://raw.githubusercontent.com/nltk/nltk_data/gh-pages/packages/corpora/twitter_samples.zip

**Hierarchical Graph Attention Network (HGAT)** (Yang et al., 2021): HGAT is a method that leverages additional information from open knowledge bases and their relationships to address the issue of semantic sparsity in short text classification.

**BERT** (Devlin et al., 2019): The BERT model, leveraging a pre-trained bidirectional Transformer architecture, captures contextual information in text and is widely employed in short text classification, effectively addressing challenges such as semantic ambiguity and feature sparsity.

**Pattern Exploiting Training (PET)** (Schick & Schütze, 2021): PET is a few-shot learning method that enhances model performance by designing task-specific prompts, allowing the model to identify patterns from a limited number of examples. This approach proves particularly effective in natural language processing tasks with scarce labeled data.

**Unified Prompt Tuning (UPT)** (Zang et al., 2022): The method explicitly captures universal prompt semantics from non-target datasets. It introduces a unified paradigm, PromptOptions-Verbalizer, for joint prompt-tuning and proposes an auxiliary knowledge-enhanced selective Masked Language Model(MLM) to capture task-agnostic prompt knowledge.

**Regular Prompt-tuning (PL)** (Liu et al., 2021): The method embeds input sentences into handcrafted templates and builds the label vocabulary space exclusively using category names. For fairness, PL employs manual templates that align with our method.

**P-tuning**(Liu et al., 2023): A method that improves PLMs by optimizing continuous prompt embeddings, achieving superior task-specific performance with minimal fine-tuning.

**Knowledge Prompt Tuning (KPT)** (Hu et al., 2022): The method enhances prompt-tuning by expanding the verbalizer with an external knowledge base, incorporating external knowledge into the prediction process through its construction, refinement, and utilization.

### 4.3 Experiment Settings

In the experimental setup, K (5, 10, 20) instances are randomly selected to form the training sets, with validation sets of equivalent sizes. The remaining samples within each subset are retained as test sets. To address the limitations posed by small-scale training datasets on baseline methods, we constructed training sets of varying sizes for neural network-based and fine-tuning approaches. Specifically, for the AG's News, Snippets, TMN News, and Twitter datasets, we set the number of training samples to 1000, 2000, and 4000 for the AG's News dataset, and 800, 1600, and 3200 for the Snippets, TMN News, and Twitter datasets. These different-scale training sets are designed to assess the model's performance under varying data volumes, thereby providing a comprehensive evaluation of the robustness and effectiveness of the methods.

The BERT-base-uncased model(Devlin et al., 2019) is trained for 10 epochs with a batch size of 32. Similarly, the HGAT model employed its original parameter configuration, including a hidden dimension of 512, a two-layer graph neural network, a learning rate of 0.005, and a dropout rate of 0.8. For the PET model, training adhered to its default settings, with a learning rate of 1e-5, a batch size of 16, and a maximum sequence length of 256. To enhance training diversity across datasets, the sample size is doubled during each iteration.

Prompt-tuning models, including UPT, PL, P-Tuning, KPT, and the proposed SPIE, utilized xlm-roberta-large (Conneau et al., 2019) as the underlying pre-trained language models. Training parameters included a learning rate of 5e-5, a weight decay of 1e-5, a dropout rate of 0.5 to prevent overfitting, and a batch size of 32. Hyper-parameters are fine-tuned through validation, with the training epochs fixed at 10 to ensure comprehensive model training. The Adam optimizer is used to optimize the model weights.

The experiments were performed on a server featuring an NVIDIA GeForce RTX 3090 Founders Edition GPU, an Intel(R) Core(TM) i9-10980XE CPU (3.00 GHz), and 125 GB of system memory. The software environment comprised Python 3.9.16 and PyTorch-CUDA 11.7.

To evaluate detection performance, Accuracy, and F1-score are adopted as metrics. Accuracy (Acc) measures the proportion of correctly predicted samples out of the total number of samples, providing an overall

Table 2: The Accuracy and F1-Score results on four datasets. The bolder ones mean better.

| Shot | Method | AG's News | | Snippets | | TMN News | | Twitter | |
|------|--------|-----------|--------|----------|--------|----------|--------|---------|--------|
| | | Acc | F1-S | Acc | F1-S | Acc | F1-S | Acc | F1-S |
| 5 | HGAT | 0.6346 | 0.6304 | 0.7210 | 0.7133 | 0.6046 | 0.5980 | 0.5518 | 0.5499 |
| | BERT | 0.3230 | 0.3119 | 0.3530 | 0.3417 | 0.3131 | 0.3007 | 0.4513 | 0.4513 |
| | PET | 0.7448 | 0.7431 | 0.7254 | 0.7156 | 0.6231 | 0.6166 | 0.6960 | 0.6866 |
| | PL | 0.7427 | 0.7340 | 0.7793 | 0.7679 | 0.5692 | 0.5610 | 0.6756 | 0.6700 |
| | P-tuning | 0.7360 | 0.7356 | 0.7197 | 0.7073 | 0.5707 | 0.5723 | 0.7163 | 0.7149 |
| | UPT | 0.7556 | 0.7550 | 0.7391 | 0.7384 | 0.6796 | 0.6796 | 0.7246 | 0.7268 |
| | KPT | 0.7552 | 0.7509 | 0.8048 | 0.8017 | 0.6121 | 0.6142 | 0.7480 | 0.7379 |
| | SPIE | **0.7956** | **0.7957** | **0.8324** | **0.8304** | **0.6953** | **0.6970** | **0.7686** | **0.7655** |
| 10 | HGAT | 0.6839 | 0.6783 | 0.7649 | 0.7618 | 0.6632 | 0.6573 | 0.6796 | 0.6764 |
| | BERT | 0.6876 | 0.6883 | 0.6683 | 0.6556 | 0.6461 | 0.6462 | 0.6026 | 0.6007 |
| | PET | 0.7723 | 0.7725 | 0.7776 | 0.7584 | 0.6953 | 0.6811 | 0.7392 | 0.7396 |
| | PL | 0.7892 | 0.7903 | 0.8000 | 0.7974 | 0.7089 | 0.7074 | 0.7043 | 0.7038 |
| | P-tuning | 0.7501 | 0.7536 | 0.7662 | 0.7656 | 0.7017 | 0.7018 | 0.7086 | 0.7025 |
| | UPT | 0.7721 | 0.7747 | 0.7982 | 0.7989 | 0.6946 | 0.6908 | 0.7973 | 0.7952 |
| | KPT | 0.7959 | 0.7967 | 0.8289 | **0.8237** | 0.7275 | 0.7278 | 0.7943 | 0.7923 |
| | SPIE | **0.8297** | **0.8296** | **0.8302** | 0.8234 | **0.7421** | **0.7404** | **0.8100** | **0.8082** |
| 20 | HGAT | 0.7010 | 0.7085 | 0.8122 | 0.8142 | 0.7264 | 0.7261 | 0.7146 | 0.7145 |
| | BERT | 0.7663 | 0.7524 | 0.7765 | 0.7699 | 0.7154 | 0.6815 | 0.7680 | 0.7674 |
| | PET | 0.8072 | 0.8065 | 0.8048 | 0.7967 | 0.7392 | 0.7413 | 0.7723 | 0.7690 |
| | PL | 0.7981 | 0.7989 | 0.8622 | 0.8573 | 0.7317 | 0.7301 | 0.7593 | 0.7573 |
| | P-tuning | 0.7871 | 0.7844 | 0.8285 | 0.8284 | 0.7265 | 0.7240 | 0.7953 | 0.7929 |
| | UPT | 0.8094 | 0.8060 | 0.8345 | 0.8337 | 0.7289 | 0.7271 | 0.8704 | 0.8744 |
| | KPT | 0.8169 | 0.8166 | 0.8758 | 0.8753 | 0.7567 | 0.7566 | 0.8526 | 0.8522 |
| | SPIE | **0.8505** | **0.8503** | **0.8842** | **0.8848** | **0.7739** | **0.7734** | **0.9003** | **0.9003** |

indication of the model's performance. The F1-score(F1-S) is the harmonic mean of precision and recall, offering a more balanced evaluation, particularly in cases of imbalanced class distribution.

### 4.4 Main results

The main experimental results across all four datasets are summarized in Table 2.

Firstly, as the number of training samples increases from 5-shot to 20-shot, there is a consistent improvement in classification performance across all methods. This trend underscores the positive impact of expanding the training dataset size on the effectiveness of short text classification.

Prompt-tuning methods outperform traditional neural network models like HGAT and fine-tuning PLMs methods like BERT, even when the latter have more training samples. We believe that prompt-tuning allows for the creation of different prompts for the short text classification task, enabling the model to learn better semantic representations from limited samples.

Both our method and the KPT approach demonstrate superior performance compared to other techniques, including regular prompt-tuning, PET, P-Tuning, and UPT. This observation highlights the value of incorporating external knowledge to alleviate the sparsity issue in short text classification. Furthermore, hard-template-based prompt-tuning consistently outperforms soft-template-based prompt learning, emphasizing the advantage of manually designed prompts that integrate prior knowledge. However, the manually crafted approach is often time-intensive and laborious. In contrast, our constructed soft prompt-tuning method effectively balances these challenges by avoiding the inefficiencies of manual template creation while address-

ing the suboptimal performance of fully automated prompt generation methods. These findings confirm the practicality and robustness of the proposed approach.

In contrast to KPT, which extracts verbalizers from external knowledge graphs, our method extracts nouns (specifically, adjectives in the case of sentimental classification like Twitter) from the original training dataset. From the experimental results, it is clear that our method outperforms KPT, we believe that internal knowledge expansion maintains linguistic consistency with task text and makes more effective use of training data. Moreover, the external knowledge base may contain significant noise, while extracting knowledge from the training dataset can effectively reduce the influence of omissions and biases in noise. In particular, our method optimizes the extracted knowledge through various techniques, enhancing their accuracy. Notably, on sentimental classification tasks like Twitter, which contain numerous emotion-expressing adjectives, our method achieves an accuracy of 90.03%. This strongly demonstrates the effectiveness of the approach we proposed.

## 4.5 Ablation study

To evaluate the impact of different tools for expanding knowledge from the training dataset, a dismantling experiment is conducted. The results are detailed in Table 3. Initially, the effectiveness of extended words extracted by each tool is examined individually. Results indicate that the performance of adjectives extracted varies across datasets. For AG's News and Snippets, NLTK demonstrated the highest accuracy, whereas SpaCy outperformed other tools on TMN News and Twitter. Although TexBlob showed slightly lower performance across all four datasets, its removal led to a noticeable decline in classification accuracy. These findings suggest that TexBlob, despite its relative inferiority, still contributes positively to overall classification performance, validating its inclusion as a valuable method in the expansion process.

Table 3: The impact of different variants on the strategy integration of label word space expansion is evaluated, with experiments conducted using accuracy across four datasets.

| NLTK | Spacy | TextBlob | AG's News | Snippets | TMN News | Twitter |
|---|---|---|---|---|---|---|
| ✓ | ✗ | ✗ | 0.8236 | 0.8441 | 0.7407 | 0.8093 |
| ✗ | ✓ | ✗ | 0.7910 | 0.8372 | 0.7489 | 0.8696 |
| ✗ | ✗ | ✓ | 0.8118 | 0.8385 | 0.7231 | 0.8283 |
| ✓ | ✓ | ✗ | 0.8239 | 0.8671 | 0.7610 | 0.8709 |
| ✓ | ✗ | ✓ | 0.8319 | 0.8557 | 0.7396 | 0.8520 |
| ✗ | ✓ | ✓ | 0.8261 | 0.8416 | 0.7515 | 0.8781 |
| ✓ | ✓ | ✓ | **0.8505** | **0.8842** | **0.7739** | **0.9003** |

## 4.6 Time complexity analysis

To validate that our method outperforms the methods with less computational time, we conducted a runtime comparison between our SPIE and KPT. The specific experimental results are presented in Table 4. Notably, in the KPT, we solely select the label word and subsequently derive pertinent concepts via related words and optimization. The experimental results indicate that our method boasts significantly lower time complexity than the method based on external knowledge expansion.

Table 4: Comparison of running time between our method and the method based on external extended knowledge across four different datasets.

| Statistic | AG's News | Snippets | TMN News | Twitter |
|---|---|---|---|---|
| KPT | 1563.88 | 2075.72 | 1905.91 | 1052.76 |
| SPIE | 885.01 | 374.97 | 499.81 | 668.70 |

Table 5: Overview of the different templates used across the four datasets.

| Id | Template |
|----|----------|
| 1 | A {"mask"} news: {"placehoder": "text_a"}. |
| 2 | {"placehoder": "text_a"}, this topic is about {"mask"}. |
| 3 | The category of {"placehoder": "text_a"} is {"mask"}. |
| 4 | Topic:{"mask"}, news:{"placehoder": "text_a"}. |
| 5 | {"placehoder": "text_a"} is related to {"mask"}. |

Table 6: The 20-shot accuracy results with different templates across four datasets.

| Template_id | AG's News | Snippets | TMN News | Twitter |
|-------------|-----------|----------|----------|---------|
| 1 | 0.7951 | 0.8728 | 0.7355 | 0.8386 |
| 2 | **0.8649** | 0.8622 | 0.7735 | 0.8734 |
| 3 | 0.8185 | 0.8758 | 0.7678 | 0.8520 |
| 4 | 0.8325 | 0.8530 | **0.7832** | 0.7573 |
| 5 | 0.8163 | **0.8855** | 0.7717 | **0.9174** |
| Avg | 0.8254 | 0.8698 | 0.7663 | 0.8477 |
| Soft | 0.8505 | 0.8842 | 0.7739 | 0.9003 |

### 4.7 Influence of the Templates

To examine the influence of different templates on experimental outcomes, five hand-crafted prompt templates were designed, with their specific contents provided in Table 5. The corresponding results are summarized in Table 6. The findings demonstrate that prompt templates effectively guide the model in understanding text content, thereby improving classification performance. The most significant enhancements were observed on Twitter and AG's News datasets.

Additionally, notable variations in text style, semantics, and other characteristics across datasets underscore the limitations of generic prompt templates. This necessitates the development of dataset-specific templates to optimize performance. Consequently, soft templates are adopted for input construction. Experimental results show that, although soft templates might not always outperform certain manually crafted templates, they generally achieve better overall average performance compared to manual templates. This highlights their potential as a more adaptable and efficient approach for text classification tasks.

### 4.8 Parameter Sensitivity

Experiments are conducted to investigate the impact of different parameters, specifically learning rate and batch size, on the experimental results. The findings are illustrated in Figure 3. The learning rate, which determines the step size for parameter updates during training, exhibited varying effects across datasets. Experimental results indicate that most datasets achieve better performance under lower learning rates, likely due to smoother parameter updates that minimize the impact of noise. However, sentiment-oriented datasets, such as Twitter, demonstrated greater tolerance for higher learning rates, which facilitated faster convergence.

Batch size, defined as the number of samples processed in each training pass, influences training dynamics, memory usage, and convergence. Results suggest that increasing the batch size within a reasonable range improves model performance across all datasets. This enhancement can be attributed to the ability of larger batch sizes to reduce gradient noise, leading to more stable updates. Moreover, larger batch sizes effectively leverage the parallel processing capabilities of GPUs, further accelerating convergence and improving training efficiency.

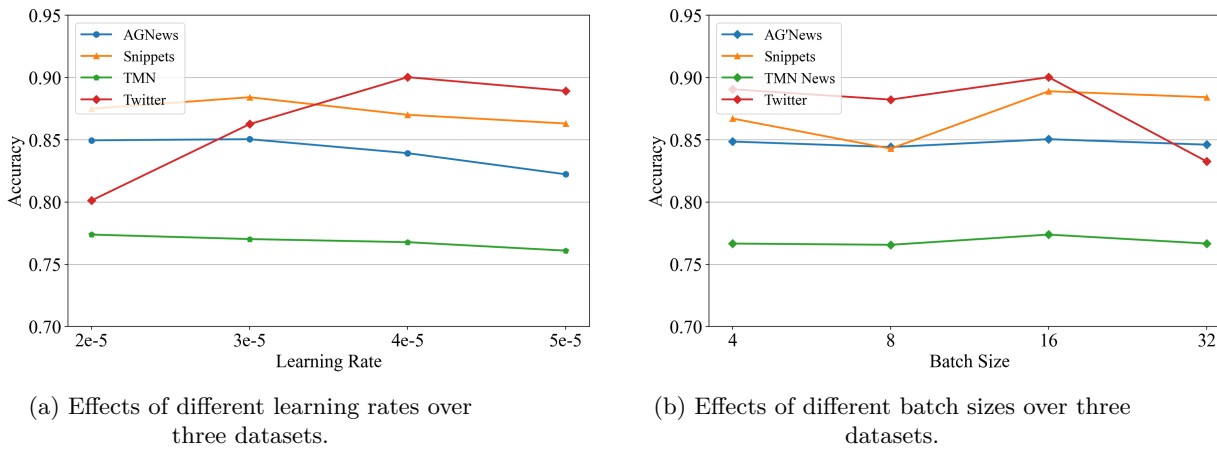

(a) Effects of different learning rates over three datasets.

(b) Effects of different batch sizes over three datasets.

Figure 3: Effects of parameter sensitivity

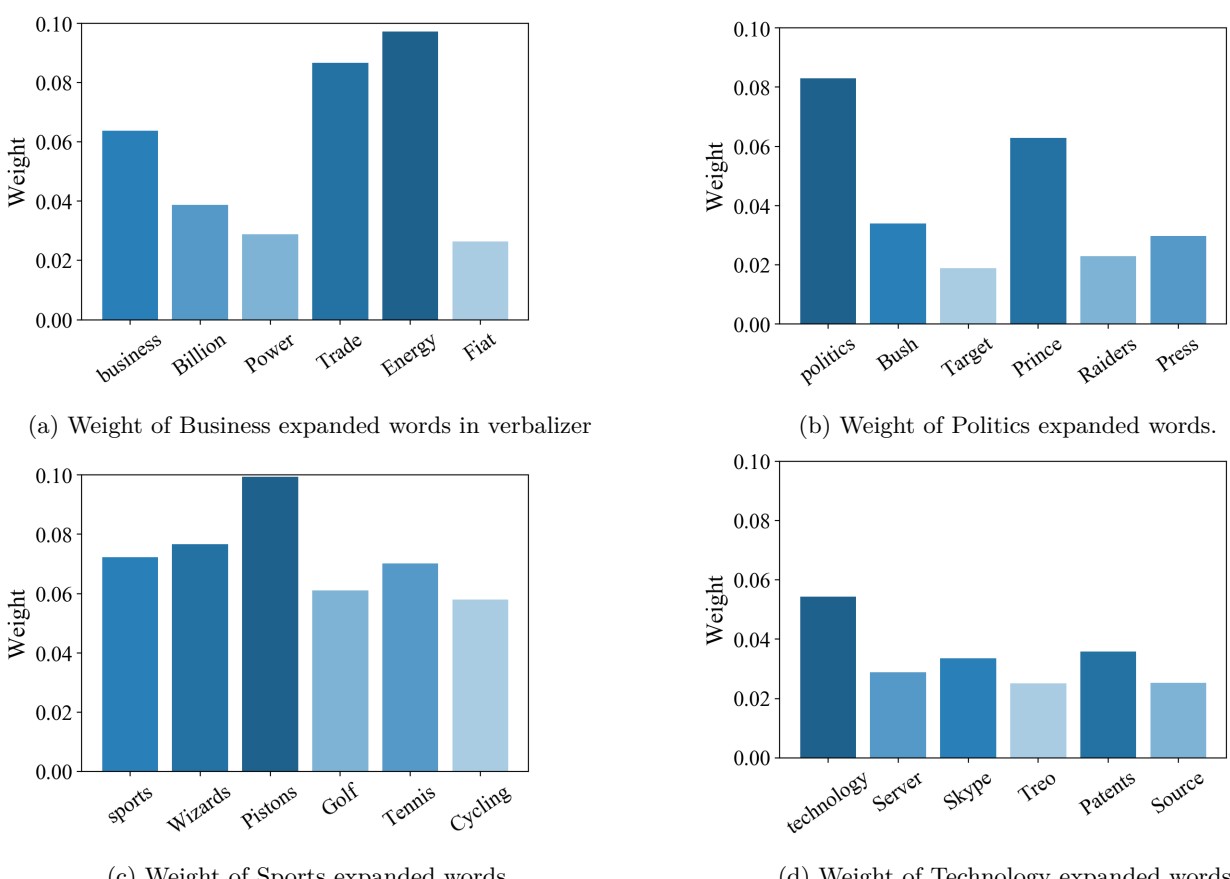

(a) Weight of Business expanded words in verbalizer

(b) Weight of Politics expanded words.

(c) Weight of Sports expanded words.

(d) Weight of Technology expanded words.

Figure 4: Weight of AG's News expanded words.

## 4.9 Visualization of expanded verbalizer weights

In order to improve the generalizability of the expansion knowledge analysis, we computed the average weight for each extended word across different datasets. For example, Figure 4 illustrates the weights of the extended words for each class during model training, as selected from the AG's News dataset. In

the "Sports" category, the word "Pistons" exhibits the highest weight, whereas "Cycling" shows the lowest weight. These results deviate from our initial expectations, suggesting that pre-trained language models may perceive and interpret words in a manner distinct from human understanding. Additionally, variations in weight distributions across different datasets further highlight the sensitivity of the model to dataset-specific characteristics.

Furthermore, The weight allocation to answer words reveals the model's sensitivity to different feature types and its learning biases. By analyzing these weights, valuable insights can be gained into the model's focus on different elements during the learning process. In future work, we plan to further investigate the influence of diverse extended vocabularies on model performance, as well as the interpretability of weight distributions. This line of research is expected to provide a deeper understanding of the underlying mechanisms that govern the model's attention and decision-making processes.

## 5 Conclusion

In this paper, we present SPIE (Soft Prompt-tuning method via Internal Knowledge Expansion), a novel method for short text classification. In contrast to the methods relying on the external knowledge base, SPIE reduces external noise and avoids manual template construction. It leverages label word expansion from the internal training dataset. Adjectives and nouns from the training set are clustered and optimized for constructing a verbalizer. Extensive experiments demonstrate the superior effectiveness of SPIE with much less computational time compared to other methods that introduced external knowledge. In future work, we aim to improve verbalizer construction and apply internal knowledge expansion to a wider range of tasks.

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
