# OpenReview forum: "Soft Prompt-tuning for Short Text Classification via Internal Knowledge Expansion"
_TMLR — Withdrawn by Authors_

### Review · Reviewer_RmRi · 2025-05-12

**Summary Of Contributions:**

This paper focuses on improving prompt-tuning for text classification. The authors propose an approach that automatically discovers effective verbalizers without relying on external resources. Their method involves identifying potential keywords through part-of-speech tagging using three different libraries, followed by clustering these keywords. By integrating these discovered verbalizers with soft prompt-tuning, the authors claim that they achieve performance improvements without relying on external resources. They provide experimental results to validate their claims.

**Audience:**

Yes

**Claims And Evidence:**

No

**Requested Changes:**

- The authors should either (1) provide the results using the full training set or (2) revise the title and introduction to emphasize the paper's focus on few-shot classification.
- Provide an explanation for the overlap with the following paper.
```
Short text classification with Soft Knowledgeable Prompt-tuning, Zhu et. al., Expert Systems with Applications, 2024
```
- I suggest the authors conduct a deeper study and analysis of the verbalizer construction, as this seems to be the novelty part of their work. The current results are not comprehensive and insightful enough.
- More carefully designed controlled experiments are needed to test the effectiveness of the proposed verbalizer. For example, what would happen if we used a simple MLP instead of BiLSTM in the proposed method? What if we added BiLSTM to other baselines? What if we only used the CLS token rather than all the contextualized representations of the soft tokens?
- Consider more baselines on verbalizer design.
- Few-shot results can be unstable. Therefore, the authors should repeat the experiments with different training examples multiple times and report the average performance and the standard deviation.
- Duplicated citations:
```
Shengding Hu, Ning Ding, Huadong Wang, Zhiyuan Liu, Juan-Zi Li, and Maosong Sun. Knowledgeable prompt-tuning: Incorporating knowledge into prompt verbalizer for text classification. arXiv preprint, abs/2108.02035, 2021.
```

```
Shengding Hu, Ning Ding, Huadong Wang, Zhiyuan Liu, Jingang Wang, Juanzi Li, Wei Wu, and Maosong Sun. Knowledgeable prompt-tuning: Incorporating knowledge into prompt verbalizer for text classification. In Proceedings of the 60th Annual Meeting of the Association for Computational Linguistics (Volume 1: Long Papers), pp. 2225–2240, Dublin, Ireland, May 2022. Association for Computational Linguistics
```

**Strengths And Weaknesses:**

Weaknesses
- It seems a bit strange that the paper mentions the following, but the experimental results only show few-shot results. Is there a specific reason for this?
```
Specifically, for the AG’s News, Snippets, TMN News, and Twitter datasets, we set the number of training samples to 1000, 2000, and 4000 for the AG’s News dataset, and 800, 1600, and 3200 for the Snippets, TMN News, and Twitter datasets.
```
- In the abstract and introduction, the authors claim that part of this work's novelty comes from the "proposed" soft prompting and soft template construction. However, the soft prompt construction in their method (Section 3.3) significantly overlaps with the following published work without citation. I believe the authors should provide an explanation for this overlap. Given this, the novelty of this work appears to seems solely on the proposed verbalizer construction, which seems limited.
```
Short text classification with Soft Knowledgeable Prompt-tuning, Zhu et. al., Expert Systems with Applications, 2024
```
- The experimental comparison seems unfair. Compared to the baselines, the proposed method requires more trainable parameters, including the additional BiLSTM. Additionally, the hyper-parameters of all methods should be provided, such as the length of soft prompt tokens and the number of trainable parameters.
- All the baselines are relatively old and most them are not for verbalizer design. The authors should consider the newer works discussed in the related work, especially those on verbalizer design. I have listed two relevant works on verbalizer design as reference.
```
Automatic Multi-Label Prompting: Simple and Interpretable Few-Shot Classification, NAACL 2022
Prototypical Verbalizer for Prompt-based Few-shot Tuning, ACL 2022
```
- Few-shot results can be unstable. Therefore, the authors should repeat the experiments with different training examples multiple times and report the average performance and the standard deviation.

---

### Review · Reviewer_yXfd · 2025-05-30

**Summary Of Contributions:**

The paper presented a soft-prompting via internal knowledge extraction (SPIE) method for short text classification, which introduced internal knowledge expansion through soft prompt-tuning to address challenges in short text classification. Experimental results of a few benchmarks demonstrated the effectiveness of the SPIE.

However, the proposed method is very similar to two existing works: Soft Prompt-tuning with Self-Resource Verbalizer for short text streams (https://www.sciencedirect.com/science/article/abs/pii/S0952197624017470) and Short text classification with Soft Knowledgeable Prompt-tuning (https://www.sciencedirect.com/science/article/abs/pii/S0957417424001131). The authors need to explain the differences and novelty of this work compared to these 2 papers.

**Audience:**

Yes

**Broader Impact Concerns:**

This work is at risk of plagiarism if the authors cannot introduce their exclusive contributions compared to the above two similar works.

**Claims And Evidence:**

No

**Requested Changes:**

1. This paper should highlight unique aspects such as specific clustering strategies, BERT-based optimization, or hybrid use of POS tagging tools. If the presented SPIE builds upon earlier work, it should be explicitly framed as an extension rather than an entirely new method.

2. The authors should elaborate on the theoretical foundation of internal knowledge expansion and its advantages over external knowledge. They also need to discuss the limitations of external knowledge integration more thoroughly, especially in terms of noise and bias.

3. The paper can provide theoretical or empirical justification for selecting hierarchical clustering over alternatives (e.g., k-means), and include ablation studies comparing different clustering algorithms. It’s also necessary to clarify the role of BiLSTM in the architecture. And compare performance with/without BiLSTM.

4. For other requested changes, please refer to the above Weaknesses.

**Strengths And Weaknesses:**

**Strengths**
1. The experiments show that the proposed SPIE (unsure originality) outperforms external knowledge-based methods like KPT on several datasets while significantly reducing computational time.

**Weaknesses**
1. There is substantial overlap between SPIE and existing methods such as SKP[1] and SPSV[2], both of which also use soft prompts and internal knowledge expansion. The authors do not sufficiently distinguish their approach from these prior works.

 - [1] Short text classification with Soft Knowledgeable Prompt-tuning.

 - [2] Soft Prompt-tuning with Self-Resource Verbalizer for short text streams.

2. While the paper mentions using hierarchical clustering (HC) to group extracted nouns/adjectives, the choice of HC over other clustering algorithms (e.g., k-means, DBSCAN) is not justified. The rationale for selecting cluster centers (label words) and the impact of hyperparameters (e.g., distance metrics, linkage criteria) are left unexplained.

3. The "BERT Prediction strategy" for optimizing label words lacks technical details. For instance, how are the top-N words selected based on [MASK] probabilities? Is this a static selection or dynamic during training? The absence of pseudocode or mathematical formulation weakens reproducibility.

4. The use of BiLSTM for contextual updates after the PLM encoder is poorly motivated. Why not use the PLM’s native attention mechanisms? The paper fails to explain how BiLSTM contributes to the model’s performance, raising questions about architectural necessity.

5. The employment of BERT for verbalizer optimization could introduce overfitting, especially in low-data regimes. The paper does not report metrics like validation loss curves or early-stopping criteria, leaving concerns about model stability unaddressed.

6. While SPIE outperforms KPT, the comparison is skewed. KPT relies on external knowledge, whereas SPIE uses internal knowledge. The paper should include ablation experiments where SPIE is modified to use external knowledge (e.g., replacing internal words with ConceptNet terms) to isolate the impact of the knowledge source.

7. The intersection of POS tags from the tools assumes they are equally reliable, but the paper does not validate this assumption. For example, do SpaCy and NLTK produce conflicting tags for slang terms in Twitter data? A confusion matrix or inter-annotator agreement metric (e.g., Cohen’s kappa) would strengthen this step.

8. There are no other pre-trained models except BERT employed in this paper, on the one hand, BERT is not the best model compared to the current large language models (LLMs), on the other hand, given the superiority of the LLMs, what is the meaning of this work?

---

### Review · Reviewer_7YF7 · 2025-08-01

**Summary Of Contributions:**

I reviewed the paper mentioned by Reviewer RmRi. These two papers are very similar including the writing. We do not know if they are from the same authors. So please investigate it. Thanks.

**Audience:**

No

**Broader Impact Concerns:**

Too similar to another paper

**Claims And Evidence:**

Yes

**Requested Changes:**

None

**Strengths And Weaknesses:**

waiting for the opinions of AC

---

### Note · Authors · 2025-08-02

**Comment:**

Dear Dr. Aditya Menon,

We are writing to formally request the withdrawal of our manuscript entitled "Soft Prompt-tuning for Short Text Classification via Internal Knowledge Expansion" (Manuscript ID: 4645), which we submitted to TMLR on April 10, 2025.

During the review process, a reviewer raised concerns that our submitted manuscript bears significant similarity to our previously published papers. After thorough internal discussions among all co-authors, we carefully reviewed the manuscript and concluded that substantial revision is required to address these concerns and to ensure the originality and integrity of our work.

Therefore, we respectfully request to withdraw this manuscript from consideration for publication in TMLR. We sincerely apologize for any inconvenience this may cause to the journal, the reviewers, and the editorial office. We deeply appreciate the time and effort that the reviewers and editors have already devoted to our manuscript.

Thank you very much for your understanding.

Sincerely,

Ye Wang, Yi Zhu, Yun Li, Jipeng Qiang, Yunhao Yuan

On behalf of all authors

**Withdrawal Confirmation:**

I have read and agree with the venue's withdrawal policy on behalf of myself and my co-authors.